# Expression of EMP1, EMP2, and EMP3 in breast phyllodes tumors

**Yoon Jin Cha**[iD]**, Ja Seung Koo**[iD]*

Department of Pathology, Yonsei University College of Medicine, Seoul, South Korea

* kjs1976@yuhs.ac

## Abstract

### Purpose

Phyllodes tumors (PTs) are biphasic tumors accounting for 0.3–1.5% of all breast tumors. Epithelial membrane proteins (EMPs) have been reported in various malignant tumors but their expression in PTs is unclear. In this study, we aimed to evaluate the expression of EMP1, EMP2, and EMP3 in breast phyllodes tumors (PTs), and to investigate their clinical implications.

### Methods

In total, 185 PTs were used for constructing a tissue microarray. Immunohistochemical staining for EMP1, EMP2, and EMP3 was performed, and the results were analyzed along with the clinicopathologic parameters.

### Results

In total, 185 PTs were included in this study, and comprised 138 benign, 32 borderline, and 15 malignant PTs. In malignant PTs, the epithelial component showed decreased expression of EMP1 ($P = 0.027$), EMP2 ($P = 0.004$), and EMP3 ($P = 0.032$), compared to the benign and borderline PTs. Conversely, stromal component of borderline and malignant PTs showed higher expression of EMP1 ($P = 0.027$), EMP2 ($P = 0.004$), and EMP3 ($P = 0.032$) compared to benign PTs. Expression of EMP1 and EMP3 correlated positively with stromal cellularity and cellular atypia ($P < 0.001$). In the univariate analysis, stromal EMP3 was associated with shorter disease-free survival ($P < 0.001$), and shorter overall survival ($P = 0.034$).

### Conclusion

The expression of EMP1, EMP2, and EMP3 is decreased in the epithelial component and is increased in the stromal component of PT with higher histologic grade. Thus, stromal EMP3 expression may serve as an independent prognostic factor in PT.

**Data Availability Statement:** All relevant data are within the paper and its Supporting Information files.

**Funding:** YJC received faculty research grant from the Yonsei University College of Medicine (6-2018-0080). The funders had no role in study design,

data collection and analysis, decision to publish, or preparation of the manuscript.

**Competing interests:** The authors have declared that no competing interests exist.

# Introduction

Phyllodes tumors (PT) are biphasic tumors accounting for 0.3–1.5% of all breast tumors. The World Health Organization (WHO) classifies PTs as benign, borderline, and malignant based on the evaluation of the stromal component [1]. PTs can recur and metastasize heterogeneously [1]. Although their stromal component is considered the main neoplastic element in PT [2], epithelial-stromal interaction is also thought to be involved in PT pathogenesis. The epithelial-stromal interaction of PTs is suggested to involve the Wnt pathway [3], platelet-derived growth factor (PDGF)/PDGF receptor(R)-β pathway [4], insulin-like growth factor (IGF)-I/II [5], and C-X-C receptor type 4 (CXCR4) [6]. The MED12 mutation is also known as a driver of tumorigenesis in fibroepithelial tumors [7, 8]. Recently, two mechanisms are suggested to underlie the progression of the histologic grade of PT: fibroepithelial tumor and benign PT show frequent somatic MED12 mutation and additional genetic alterations are found with increasing histologic grade, whereas borderline/malignant PTs without MED12 mutation frequently harbor TP53 and PIK3CA mutations [9, 10].

Epithelial membrane proteins (EMPs; EMP1, EMP2, and EMP3) are members of the peripheral myelin protein (PMP22) gene family [11]. EMP1 is a target of c-MYC [12], and is highly expressed in undifferentiated cells [13]; it has been reported as a negative regulator in some cancers including nasopharyngeal cancer [14], and breast cancer [15]. EMP2 has been considered an oncogene, particularly in hormone-related cancers such as endometrial and breast cancer [16, 17]. EMP3 appears to be a tumor suppressor gene in solid tumors [18]. So far, EMPs have been evaluated in various malignant tumors, particularly, brain tumors and carcinomas. However, EMP expression in breast PTs has not been elucidated. As PT is a biphasic neoplasm, EMP expression in both epithelial and stromal components, as well as in different histologic grades, is expected to differ. In the present study, we aimed to evaluate the expression and clinical implications of EMP1, EMP2, and EMP3 in breast PTs.

# Materials and methods

## Patient selection

Tissue samples were collected from patients with a pathologically confirmed diagnosis of PT who underwent resection at the Severance hospital between 2000 and 2010. The study was approved by the Institutional review board of Yonsei university, Severance hospital, with wavier of informed consent. All clinical data were anonymized. All tissues were fixed in 10% buffered formalin and embedded in paraffin. All archival hematoxylin and eosin (H&E)-stained slides for each case were reviewed by two pathologists (JS Koo and YJ Cha), and all PTs were assigned a histologic grade based on the WHO classification [1]. Clinical factors including patient age at diagnosis, tumor recurrence, distant metastasis, and patient survival were examined.

## Tissue microarray

On H&E-stained slides of tumors, a representative area was selected, and the corresponding spot was marked on the surface of the paraffin block. Using a biopsy needle, the selected area was punched out and the resulting 5-mm tissue core was placed in a 5 × 6 recipient block. Two tissue cores were extracted from each case to minimize extraction bias. Each separate tissue core was assigned a unique tissue microarray location number that was linked to a database including other clinicopathologic data.

**Table 1. Source, clone, and dilution of the antibodies used.**

| Antibody | Company | Clone | Dilution |
|---|---|---|---|
| EMP1 | Abcam, Cambridge, UK | N-terminal | 1:100 |
| EMP2 | Abcam, Cambridge, UK | C-terminal | 1:100 |
| EMP3 | Santa Cruz Biotechnology, CA, USA | SW-5 | 1:100 |

EMP, epithelial membrane protein.

## Immunohistochemistry and interpretation

The antibodies used for immunohistochemistry in this study are shown in Table 1. All immunostaining procedures were performed using formalin-fixed, paraffin-embedded tissue sections. Briefly, 5-μm-thick sections were prepared using a microtome, transferred to adhesive

**Table 2. Clinicopathologic characteristics of patients with phyllodes tumor.**

| Parameters | Total N = 185 (%) | PT, benign N = 138 (%) | PT, borderline N = 32 (%) | PT, malignant N = 15 (%) | P-value |
|---|---|---|---|---|---|
| Age, years (mean ± SD) | 40.4 ± 12.2 | 39.1 ± 12.1 | 43.2 ± 11.0 | 47.6 ± 13.4 | 0.013 |
| Tumor size, cm (mean ± SD) | 4.0 ± 2.6 | 3.7 ± 2.2 | 4.2 ± 2.5 | 6.2 ± 4.3 | 0.001 |
| Stromal cellularity | | | | | <0.001 |
| Mild | 107 (57.8) | 105 (76.1) | 2 (6.3) | 0 (0.0) | |
| Moderate | 66 (35.7) | 33 (23.9) | 26 (81.3) | 7 (46.7) | |
| Marked | 12 (6.5) | 0 (0.0) | 4 (12.5) | 8 (53.3) | |
| Stromal atypia | | | | | <0.001 |
| Mild | 143 (77.3) | 136 (98.6) | 7 (21.9) | 0 (0.0) | |
| Moderate | 32 (17.3) | 2 (1.4) | 22 (68.8) | 8 (53.3) | |
| Marked | 10 (5.4) | 0 (0.0) | 3 (9.4) | 7 (46.7) | |
| Stromal mitosis (per 10 HPFs) | | | | | <0.001 |
| 0–4 | 142 (76.8) | 138 (100.0) | 4 (12.5) | 0 (0.0) | |
| 5–9 | 33 (17.8) | 0 (0.0) | 28 (87.5) | 5 (33.3) | |
| ≥ 10 | 10 (5.4) | 0 (0.0) | 0 (0.0) | 10 (66.7) | |
| Stromal overgrowth | | | | | <0.001 |
| Absent | 169 (91.4) | 138 (100.0) | 29 (90.6) | 2 (13.3) | |
| Present | 16 (8.6) | 0 (0.0) | 3 (9.4) | 13 (86.7) | |
| Tumor margin | | | | | <0.001 |
| Circumscribed | 166 (89.7) | 135 (97.8) | 25 (78.1) | 6 (40.0) | |
| Infiltrative | 19 (10.3) | 3 (2.2) | 7 (21.9) | 9 (60.0) | |
| Surgical procedure | | | | | <0.001 |
| Local excision | 136 (73.5) | 119 (86.2) | 16 (50.0) | 1 (6.7) | |
| Wide excision | 38 (20.5) | 14 (10.1) | 15 (46.9) | 9 (60.0) | |
| Mastectomy | 11 (5.9) | 5 (3.6) | 1 (3.1) | 5 (33.3) | |
| Margin status | | | | | 0.928 |
| Negative | 160 (86.5) | 120 (87.0) | 27 (84.4) | 13 (86.7) | |
| Positive | 25 (13.5) | 18 (13.0) | 5 (15.6) | 2 (13.3) | |
| Tumor local recurrence | 17 (9.2) | 5 (3.6) | 5 (15.6) | 7 (46.7) | <0.001 |
| Distance metastasis | 7 (3.8) | 0 (0.0) | 0 (0.0) | 7 (46.7) | <0.001 |
| Follow-up, months (median, range) | 63 (8–183) | 73 (14–183) | 59 (12–144) | 15 (8–62) | <0.001 |

PT, phyllodes tumor; SD, standard deviation; HPFs, high power fields.

slides, and dried at 62°C for 30 minutes. After incubation with primary antibodies, immuno-detection was performed with biotinylated anti-mouse immunoglobulin, followed by peroxi-dase-labeled streptavidin using a labeled streptavidin biotin kit with 3,3′-diaminobenzidine as the chromogenic substrate. Appropriate positive and negative controls were included. Slides were counterstained with Harris hematoxylin. The staining of all immunohistochemical markers was assessed by light microscopy and samples were scored by multiplying the proportion of stained cells (0%, negative; 1, <30% positivity, 2; ≥30% positivity) with the staining intensity (0, negative; 1, weak; 2, moderate; 3, strong). Representative pictures of staining is shown in S1 and S2 Figs. Multiplied values of 0 and 1 were considered as negative whereas values of 2 or more were considered as positive [19].

## Statistical analysis

Data were analyzed using SPSS for Windows, Version 18.0 (SPSS Inc., Chicago, IL, USA). For determination of statistical significance, Student's $t$ test and Fisher's exact test were used for continuous and categorical variables, respectively. Statistical significance was considered at $P < 0.05$. Kaplan-Meier survival curves and log-rank statistics were employed to evaluate the time to tumor recurrence. Multivariate regression analysis was performed using the Cox proportional hazards model.

## Results

### Basal characteristics of PTs

Table 2 shows the basal clinical characteristics of patients. In total, 185 cases were included in this study and were composed of 138 benign, 32 borderline, and 15 malignant PTs. Increasing

**Table 3. Expression of EMP1, EMP2, and EMP3 in phyllodes tumors.**

| Parameters | Total N = 185 (%) | PT, benign N = 138 (%) | PT, borderline N = 32 (%) | PT, malignant N = 15 (%) | *P*-value |
|---|---|---|---|---|---|
| EMP1 (E)* | | | | | 0.027 |
| Negative | 9 (5.4) | 5 (3.6) | 3 (11.1) | 1 (33.3) | |
| Positive | 159 (94.6) | 133 (96.4) | 24 (88.9) | 2 (66.7) | |
| EMP1 (S) | | | | | <0.001 |
| Negative | 81 (43.8) | 75 (54.3) | 3 (9.4) | 3 (20.0) | |
| Positive | 104 (56.2) | 63 (45.7) | 29 (90.6) | 12 (80.0) | |
| EMP2 (E)* | | | | | 0.004 |
| Negative | 39 (23.2) | 32 (23.2) | 4 (14.8) | 3 (100.0) | |
| Positive | 129 (76.8) | 106 (76.8) | 23 (85.2) | 0 (0.0) | |
| EMP2 (S) | | | | | <0.001 |
| Negative | 176 (95.1) | 137 (99.3) | 26 (81.3) | 13 (86.7) | |
| Positive | 9 (4.9) | 1 (0.7) | 6 (18.8) | 2 (13.3) | |
| EMP3 (E)* | | | | | 0.032 |
| Negative | 24 (14.3) | 18 (13.0) | 4 (14.8) | 2 (66.7) | |
| Positive | 144 (85.7) | 120 (87.0) | 23 (85.2) | 1 (33.3) | |
| EMP3 (S) | | | | | <0.001 |
| Negative | 137 (74.1) | 118 (85.5) | 13 (40.6) | 6 (40.0) | |
| Positive | 48 (25.9) | 20 (14.5) | 19 (59.4) | 9 (60.0) | |

*Seventeen tumors without an epithelial component were excluded.

PT, phyllodes tumor; EMP, epithelial membrane protein; E, epithelial staining; S, stromal staining.

age and tumor size were associated with the histologic grade of PT ($P = 0.013$, and $P = 0.001$, respectively). Tumor recurrence and distant metastasis were more frequent with higher histologic grade ($P < 0.001$). Seven PTs showed distant metastasis, and the metastatic site for all cases was the lung (Table 2).

## EMP1, EMP2, and EMP3 expression according to the PT grades

The expression of EMP1, EMP2, and EMP3 in both the epithelial and stromal components differed according to the histologic grade (Table 3). EMP1 ($P = 0.027$), EMP2 ($P = 0.004$), and EMP3 ($P = 0.032$) expression in the epithelial component showed an inverse correlation with the histologic grade. In contrast, EMP1 ($P = 0.027$), EMP2 ($P = 0.004$), and EMP3 ($P = 0.032$) expression in the stromal component was higher in borderline and malignant PTs compared to that in benign PTs (Fig 1).

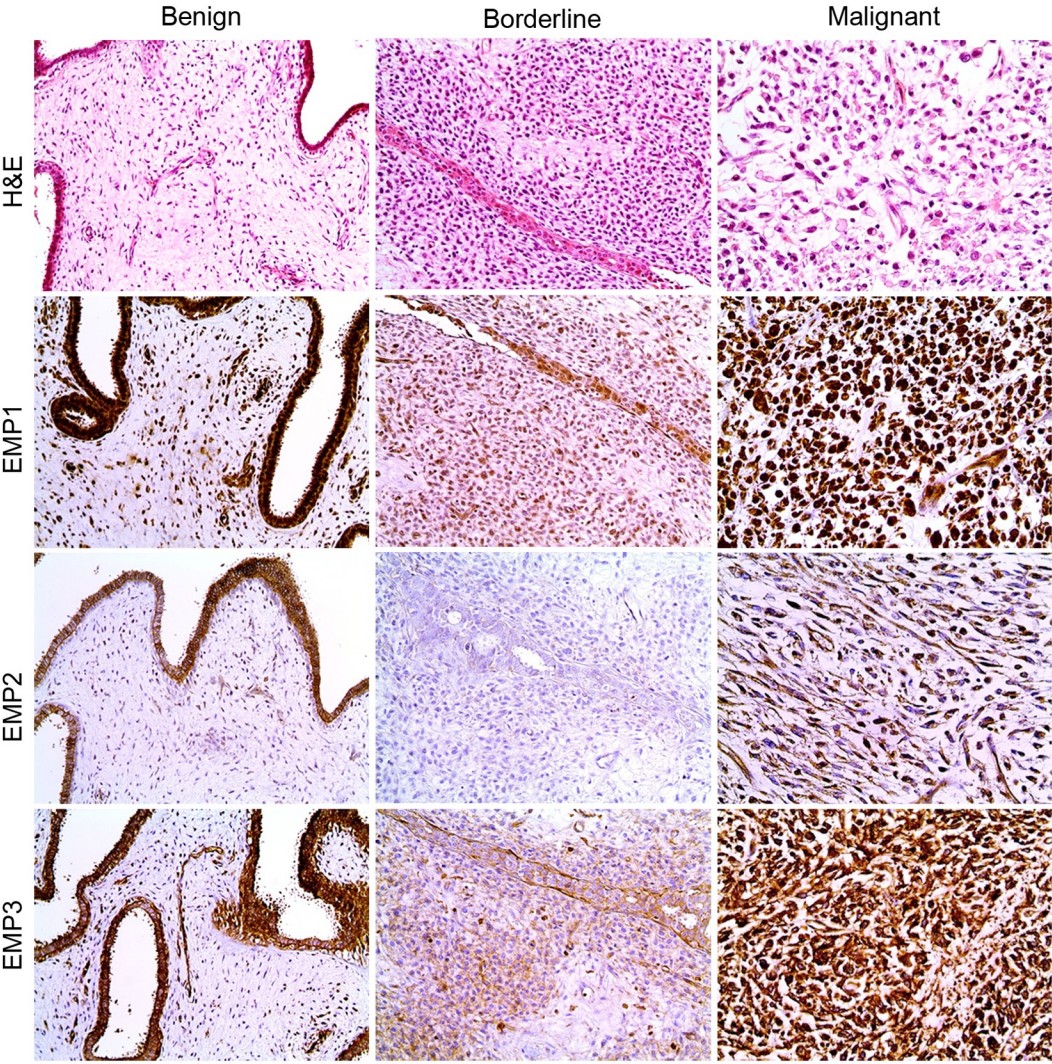

**Fig 1. Representative histologic images of hematoxylin and eosin staining and immunohistochemical staining for EMP1, EMP2, and EMP3 in phyllodes tumors with different histologic grades.** The expression of EMPs is the strongest in the epithelial component of benign phyllodes tumors (PT). Notably, strong stromal expression of EMP1, EMP2, and EMP3 is observed in malignant PT.

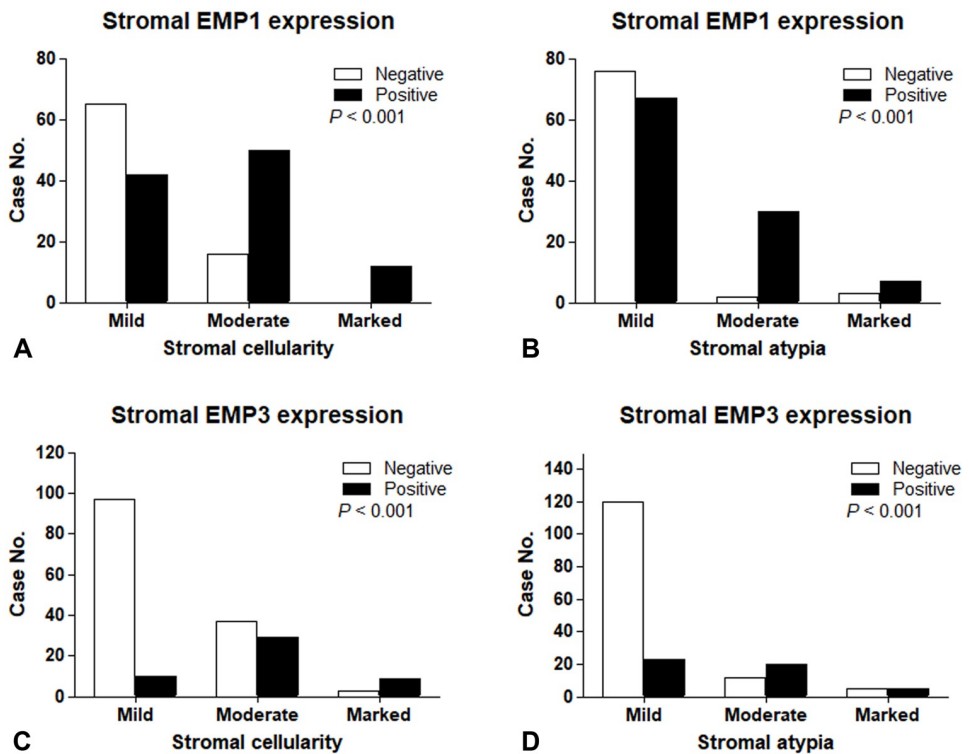

**Fig 2. Association of histology and the expression of EMP1 and EMP3.** Increased stromal cellularity and stromal atypia are correlated with the expression rate of EMP1 (A and B) and EMP3 (C and D) S, stromal.

## Correlation between EMP1, EMP2, and EMP3 expression in PTs and pathologic parameters

Stromal positivity of EMP1 and EMP3 was associated with stromal cellularity and stromal cell atypia. EMP1 expression was positively correlated with increasing stromal cellularity and cellular atypia ($P < 0.001$, Fig 2).

## Impact of EMP1, EMP2, and EMP3 expression on patient prognosis

In univariate analysis, stromal EMP3 expression was associated with shorter disease-free survival ($P < 0.001$) and shorter overall survival (OS) ($P = 0.034$) (Table 4, Fig 3). However, no significant difference for stromal EMP3 expression was found by multivariate Cox analysis (Table 5).

## Discussion

We evaluated the expression of EMP1, EMP2, and EMP3 in PTs of the breast, and found that EMP expression was reduced in the epithelial component and was increased in the stromal component, along with increasing histologic grade. Although the epithelial component showed a different expression pattern–an inverse correlation with stromal expression–we focused on the stromal component in the present study because the stromal component is the neoplastic element and determines the diagnosis and tumor grade. Although PTs account for a far lesser proportion of breast fibroepithelial lesions compared to fibroadenomas, both lesions

**Table 4. Univariate analysis of the impact of EMP1, EMP2, and EMP3 expression in phyllodes tumors.**

| Parameters | No. of patients (%) Total/recurrence/metastasis | Disease-free survival | | Overall survival | |
|---|---|---|---|---|---|
| | | Median months (range) | P-value | Median months (range) | P -value |
| EMP1 (E)* | | | N/A | | N/A |
| Negative | 9 (100.0) / 0 (0.0) / 0 (0.0) | N/A | | N/A | |
| Positive | 159 (100.0) / 10 (6.3) /1 (0.6) | N/A | | N/A | |
| EMP1 (S) | | | 0.364 | | N/A |
| Negative | 81 (100.0) / 6 (7.4) / 0 (0.0) | 166 (156–176) | | N/A | |
| Positive | 104 (100.0) / 11 (10.6) / 7(6.7) | 162 (151–174) | | N/A | |
| EMP2 (E)* | | | 0.642 | | N/A |
| Negative | 39 (100.0) / 2 (5.1) / 0 (0.0) | 169 (158–179) | | N/A | |
| Positive | 129 (100.0) /8 (6.2) / 1 (0.8) | 171 (163–179) | | N/A | |
| EMP2 (S) | | | N/A | | N/A |
| Negative | 176 (100.0) / 17 (9.7) / 7 (4.0) | N/A | | N/A | |
| Positive | 9 (100.0) / 0 (0.0) / 0 (0.0) | N/A | | N/A | |
| EMP3 (E)* | | | 0.687 | | N/A |
| Negative | 24 (100.0) / 2 (8.3) / 0 (0.0) | 134 (119–148) | | N/A | |
| Positive | 144 (100.0) / 8 (5.6) /1 (0.7) | 172 (165–179) | | N/A | |
| EMP3 (S) | | | <0.001 | | 0.034 |
| Negative | 137 (100.0) / 7 (5.1) / 3 (2.2) | 173 (167–180) | | 179 (174–183) | |
| Positive | 48 (100.0) / 10 (20.8) / 4 (8.3) | 138 (116–159) | | 163 (150–176) | |

*Seventeen tumors without an epithelial component were excluded.

PT, phyllodes tumor; EMP, epithelial membrane protein; E, epithelial staining; S, stromal staining.

share histomorphological features [20, 21], as well as genetic alterations such as recurrent MED12 mutations [7, 22–24].

In the present study, stromal EMPs showed significantly increased expression in borderline/malignant PTs, but only stromal EMP3 expression was identified as an independent risk factor for short OS. Considering that EMP1, EMP2, and EMP3 have been reported to play important roles in various malignant tumors [25], increased expression of EMP1 and EMP3, along with stromal cellularity and stromal atypia, imply that increased EMP expression in PT could suggest a higher malignant potential for PT. EMP1 also showed a tendency for increased

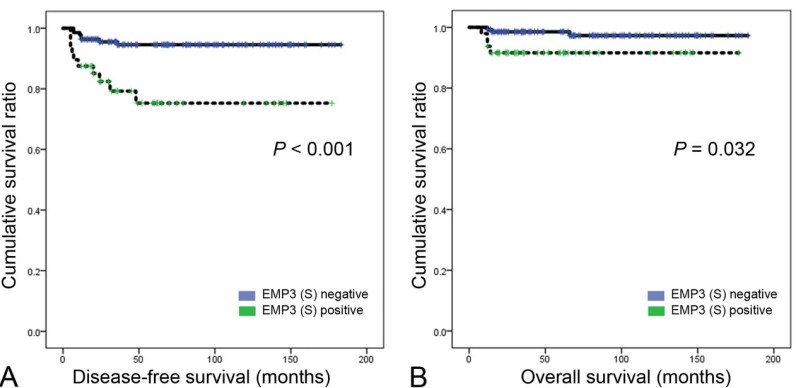

**Fig 3. Disease-free survival and overall survival based on EMP3 expression.** Cases with stromal EMP3 expression show inferior prognosis with regard to disease-free survival (A) and overall survival (B). S, stromal.

**Table 5. Multivariate Cox regression analysis of disease-free and overall survival in patients with phyllodes tumors.**

| Included factor | Disease-free survival | | Overall survival | |
|---|---|---|---|---|
| | HR (95% CI) | *P*-value | HR (95% CI) | *P*-value |
| Histologic grade | | | | |
| Benign | Reference | | Reference | |
| Borderline/malignant | 2.435 (0.536–11.060) | 0.249 | 206.6 (3.929–10866) | 0.008 |
| Stromal cellularity | | | | |
| Mild | Reference | | Reference | |
| Moderate/marked | 1.198 (0.159–9.032) | 0.861 | 0.002 (0.000–8.503) | 0.910 |
| Stromal atypia | | | | |
| Mild | Reference | | Reference | |
| Moderate/marked | 0.800 (0.111–5.774) | 0.825 | 0.000 (0.000–6.754) | 0.881 |
| Stromal mitosis | | | | |
| 0-4/10 HPFs | Reference | | Reference | |
| >4/10 HPFs | 9.550 (0.781–116.7) | 0.077 | 16125 (0.000–6.538) | 0.857 |
| Stromal overgrowth | | | | |
| Absent | Reference | | Reference | |
| Present | 3.535 (0.830–15.060) | 0.088 | 30617 (0.000–1.456) | 0.862 |
| Tumor margin | | | | |
| Circumscribed | Reference | | Reference | |
| Infiltrative | 0.558 (0.159–1.957) | 0.362 | 0.150 (0.013–1.715) | 0.127 |
| EMP3 (S) | | | | |
| Negative | Reference | | Reference | |
| Positive | 0.523 (0.153–1.787) | 0.301 | 1.841 (0.035–98.090) | 0.763 |

HR, hazard ratio; CI, confidence interval; HPFs, high power fields; EMP, epithelial membrane protein; S, stromal staining.

expression in the stroma along with an increase in the histologic grade, but did not impact prognosis. Conversely, stromal EMP2 expression was only found in a few cases (N = 9), no further statistical meaning could be found.

A previous study has shown that EMP3 is hypermethylated in approximately 20–40% of neuroblastoma and glioma cases, and plays a role in tumor suppression, which is also related with patients' prognosis [26]. As most previous studies regarding EMPs had used epithelial carcinoma and a few had used glioma, this study was important as it determined the role of EMP3 in non-epithelial tumors, similar to the present study. Another recent study on high-grade glioma showed high expression of EMP3, particularly in CD44-high glioblastoma [27], which refuted the result of a prior study on glioma [26]. However, CD44-high glioblastoma is different from the general cases of glioma; it is classified as the mesenchymal subclass within glioblastoma. EMP3 expression was found to be correlated with the activation of TGF-β/Smad2/3 signaling by interaction with TGFBR2, which resulted in TGF-β stimulated gene expression and tumor cell proliferation [27]. TGF-β signaling generally enhances epithelial mesenchymal transition (EMT) [28, 29], but it also activates the proliferation of tumor cells of non-epithelial origin [30, 31]. In gastric cancer, EMP3 has been suggested as a downstream effector of TWIST1/2 and a regulator of EMT [32].

Moreover, a previous study showed that malignant PT was more likely to have wild-type MED12 along with mutations in PIK3CA, which is considered an oncogene [9]. EMP3 and EMP1 have been reported to be involved in the PI3K/Akt pathway in HCC [33], and in the tumorigenesis of non-small cell lung cancer [34]. Because research regarding the treatment of

PT is still limited and unclear, mining of effective therapeutic targets is necessary [35, 36]. As stromal EMP3 expression showed increased expression along with the histologic grade as well as was intimately associated with tumor aggressiveness and prognosis in the present study, it might be considered as a good candidate for treatment. Moreover, EMP1 and EMP2, which also showed increased expression in borderline/malignant PT, should be also evaluated further, even though they showed no significant clinical impact in the present study. In the present study, EMP2-expressing PTs were too few in number, and were inappropriate for statistical analysis. However, EMP2 has been reported to be highly expressed in glioblastoma and in human samples and a mouse model; further, the anti-EMP antibody showed efficacy in tumor inhibition [37]. Another limitation of the present study is that there is no data evaluates the EMPs in mesenchymal tumors, probably EMPs are basically epithelial membrane proteins, as their names. As anti EMP2 antibody could affect the tumor inhibition of glioblastoma, further evaluation and validation of EMPs expression in high grade mesenchymal tumors are required.

In conclusion, the results of this study indicate that stromal expression of EMP1, EMP2, and EMP3 is increased along with the histologic grade in PT, and that stromal EMP3 expression is an independent prognostic factor for the survival of patients with breast PTs.

## Supporting information

**S1 Fig. Scan power view of all immunohistochemistry slides of EMP1, EMP2, and EMP3.** (TIF)

**S2 Fig. Higher magnification of immunohistochemistry of EMP1, EMP2, and EMP3.** (TIF)

## Author Contributions

**Conceptualization:** Yoon Jin Cha, Ja Seung Koo.

**Formal analysis:** Yoon Jin Cha.

**Investigation:** Yoon Jin Cha.

**Methodology:** Yoon Jin Cha, Ja Seung Koo.

**Project administration:** Ja Seung Koo.

**Supervision:** Ja Seung Koo.

**Visualization:** Yoon Jin Cha.

**Writing – original draft:** Yoon Jin Cha, Ja Seung Koo.

**Writing – review & editing:** Yoon Jin Cha, Ja Seung Koo.

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
