## [Decision Letter · Decision Letter 0]

2 Jun 2020

PONE-D-20-11320

Expression of EMP1, EMP2, and EMP3 in breast phyllodes tumors

PLOS ONE

Dear Dr. Koo,

Thank you for submitting your manuscript to PLOS ONE. After careful consideration, we feel that it has merit but does not fully meet PLOS ONE’s publication criteria as it currently stands. Therefore, we invite you to submit a revised version of the manuscript that addresses the points raised during the review process.

Experts have reviewed the current manuscript and found that the study has been clear and interesting but acouple of reviewers got impression for the study to be revised to make more clear stody enough to be published in the journal, with more explanation or experimental work.

We look forward to receiving your revised manuscript.

Kind regards,

Jung Weon Lee, Ph.D.

Academic Editor

PLOS ONE

Journal Requirements:

2. In your ethics statement in the manuscript and in the online submission form, please provide additional information about the patient records used in your retrospective study.

Specifically, please ensure that you have discussed whether all data were fully anonymized before you accessed them.

3. At this time, we ask that you please provide scale bars on the microscopy images presented in Figure 1 and refer to the scale bar in the corresponding Figure legend.

Reviewers' comments:

Reviewer's Responses to Questions

**Comments to the Author**

1. Is the manuscript technically sound, and do the data support the conclusions?

Reviewer #1: Partly

Reviewer #2: Yes

Reviewer #3: Yes

Reviewer #4: Yes

2. Has the statistical analysis been performed appropriately and rigorously? 

Reviewer #1: No

Reviewer #2: Yes

Reviewer #3: Yes

Reviewer #4: Yes

3. Have the authors made all data underlying the findings in their manuscript fully available?

Reviewer #1: Yes

Reviewer #2: Yes

Reviewer #3: Yes

Reviewer #4: Yes

4. Is the manuscript presented in an intelligible fashion and written in standard English?

Reviewer #1: Yes

Reviewer #2: Yes

Reviewer #3: Yes

Reviewer #4: Yes

5. Review Comments to the Author

Reviewer #1: 1. Were the pathologists blind to the identity of the patients, their clinic-pathological diagnoses, and survival data?

2. Please provide lower magnification images of all the IHC slides – demonstrating the distribution of EMP staining in the stromal vs epithelial areas.

3. Please provide higher magnification images of all the IHC slides – demonstrating cell membranous localization of the EMP proteins.

4. How was the “proportion of stained cells” calculated? How many areas per tissue were stained for histo-pathological evaluation? How many images were taken for each of the tissue sections? How was “staining intensity” determined? On what basis did the authors use “30%” as the cutoff to determine positivity of EMP staining? Please explain with appropriate references.

Reviewer #2: Authors investigated the relationship between the expression of EMP1/2/3 and clinicopathological features of phyllodes tumors, and showed EMP expression is reduced in the epithelial region, while it is increase in the stromal region, along with increasing histologic grade.

<major comments="">

1) Immunohistochemical staining is quite essential for this study, and its procedure must be standard to obtain solid data. Authors did not use any first antibodies for control. They should use the same class of non-specific antibodies to check non-specific staining, otherwise they could not tell any difference correctly in their staining.

2) Histologic grade (mitosis, atypia, cellularity) is generally thought to be the most reliable variable for phyllodes tumors. Although stromal EMP3 is significantly correlated with prognosis in univariate analysis, stromal EMP3 is not significant in multivariate Cox regression analysis, but histologic grade. This data might lead us to the conclusion that EMPs is not necessarily required for the diagnosis of phyllodes tumor.

<minor comment="">

1) Authors describe about the tissue microarray in materials and methods, but no data was found using tissue microarray. Why?

Reviewer #3: The manuscript entitled ‘Expression of EMP1, EMP2, and EMP3 in breast phyllodes tumors Yoon Jin Cha and Ja Seung Koo is a descriptive paper aimed to evaluate the expression of Epithelial membrane proteins (EMPs) EMP1, EMP2, and EMP3 in breast phyllodes tumors, and to investigate their clinical implications. This manuscript demonstrates using EMP3 expression may serve as an independent prognostic factor in phyllodes tumors. They used 185 samples in this study, with 138 benign, 32 borderline, and 15 malignant phyllodes tumors.

The subject is interesting and significant, the approach is correct. This is a generally well-written and easy to follow. The tables and figures presented in this manuscript are clear.

Reviewer #4: This is a clear, concise report. It will be of value to those interested in the EMP proteins and those investigating the biology of Phyllodes tumors. It is rare to see a report ready for publication on first submission, but this meets all the requirements for publication in PLOS One.

 </minor></major>

6. PLOS authors have the option to publish the peer review history of their article (what does this mean?). If published, this will include your full peer review and any attached files.

Reviewer #1: No

Reviewer #2: No

Reviewer #3: No

Reviewer #4: No

---

## [Author Response · Author response to Decision Letter 0]

20 Jul 2020

Reviewer #1: 

Thank you for your valuable comments. We revised the manuscript according to your helpful suggestion. 

1. Were the pathologists blind to the identity of the patients, their clinic-pathological diagnoses, and survival data?

[Answer] Yes

2. Please provide lower magnification images of all the IHC slides – demonstrating the distribution of EMP staining in the stromal vs epithelial areas.

[Answer] We provide the scan power view of all IHC slides as supplementary figure 1. (page 5, line 2-3)

3. Please provide higher magnification images of all the IHC slides – demonstrating cell membranous localization of the EMP proteins.

[Answer] As there are many EMP positive cases, we provide the representative positive slides with high magnification as supplementary figure 2. (page 5, line 2-3)

4. How was the “proportion of stained cells” calculated? How many areas per tissue were stained for histo-pathological evaluation? How many images were taken for each of the tissue sections? How was “staining intensity” determined? On what basis did the authors use “30%” as the cutoff to determine positivity of EMP staining? Please explain with appropriate references.

[Answer] Proportion of stained cells was evaluated by stained area within the TMA tissue cores. In this study, we used two 5mm tissue cores per each case. Most of stained cases demonstrated homogenous staining pattern.After review of every slides of each case, most representative areas were extracted for TMA construction. Generally, phyllodes tumor smaller than 4-5cm, all tumor content is made into paraffin block and examined with HE slides. Tumors bigger than 6cm, at least one block per centimeter is submitted for diagnosis. 

As supplementary figure we submitted, we scanned every TMA slides for taking pictures. In case of high magnification, 9-10 images can be obtained per each core. 

Staining intensity is divided by three groups, and defined relatively in each antibody. Generally, in EMP2 and EMP3, weak intensity was recognizable positivity at high magnification (400x). Strong intensity was intense positive that easily recognized under low magnification (12.5x). Moderate intensity was defined between weak and strong intensity. In EMP1, expression intensity tended to be generally strong, most of the intensity was 2 or 3. 

We applied the staining scoring system from the previous study (PMID: 19762066). In PTs, EMPs were homogenously stained in most of cases, so in many cases, positivity was determined by staining intensity. 

Reviewer #2: Authors investigated the relationship between the expression of EMP1/2/3 and clinicopathological features of phyllodes tumors, and showed EMP expression is reduced in the epithelial region, while it is increase in the stromal region, along with increasing histologic grade.

Thank you for your valuable comments. We revised the manuscript according to your helpful suggestion. 

1) Immunohistochemical staining is quite essential for this study, and its procedure must be standard to obtain solid data. Authors did not use any first antibodies for control. They should use the same class of non-specific antibodies to check non-specific staining, otherwise they could not tell any difference correctly in their staining.

[Answer] We already and always used the positive and negative controls when we set up the appropriate dilution of antibody. Accordingly, we revised the method IHC section. (page 4, line 27-28)

2) Histologic grade (mitosis, atypia, cellularity) is generally thought to be the most reliable variable for phyllodes tumors. Although stromal EMP3 is significantly correlated with prognosis in univariate analysis, stromal EMP3 is not significant in multivariate Cox regression analysis, but histologic grade. This data might lead us to the conclusion that EMPs is not necessarily required for the diagnosis of phyllodes tumor.

[Answer] We absolutely agree with your comment. We also found that the clinical impact of this study and the data might be not that significant, But there was no preceding data of EMP and PT, we wanted to investigate the expression of EMPs in PTs, and found that stromal EMPs are increasingly expressed along with histologic grade of PT, and stromal EMP3 could be an prognostic factor. We hope that our result might help other investigators studying EMPs. 

1) Authors describe about the tissue microarray in materials and methods, but no data was found using tissue microarray. Why?

[Answer] All the data in result section is the IHC and statistical results using TMA 

Reviewer #3: The manuscript entitled ‘Expression of EMP1, EMP2, and EMP3 in breast phyllodes tumors Yoon Jin Cha and Ja Seung Koo is a descriptive paper aimed to evaluate the expression of Epithelial membrane proteins (EMPs) EMP1, EMP2, and EMP3 in breast phyllodes tumors, and to investigate their clinical implications. This manuscript demonstrates using EMP3 expression may serve as an independent prognostic factor in phyllodes tumors. They used 185 samples in this study, with 138 benign, 32 borderline, and 15 malignant phyllodes tumors.

The subject is interesting and significant, the approach is correct. This is a generally well-written and easy to follow. The tables and figures presented in this manuscript are clear.

[Answer] Thank you for your comments.

Reviewer #4: This is a clear, concise report. It will be of value to those interested in the EMP proteins and those investigating the biology of Phyllodes tumors. It is rare to see a report ready for publication on first submission, but this meets all the requirements for publication in PLOS One.

[Answer] Thank you for your comments.

---

## [Decision Letter · Decision Letter 1]

18 Aug 2020

Expression of EMP1, EMP2, and EMP3 in breast phyllodes tumors

PONE-D-20-11320R1

Dear Dr. Koo,

We’re pleased to inform you that your manuscript has been judged scientifically suitable for publication and will be formally accepted for publication once it meets all outstanding technical requirements.

Kind regards,

Jung Weon Lee, Ph.D.

Academic Editor

PLOS ONE

Additional Editor Comments (optional):

Reviewers' comments:

Reviewer's Responses to Questions

**Comments to the Author**

1. If the authors have adequately addressed your comments raised in a previous round of review and you feel that this manuscript is now acceptable for publication, you may indicate that here to bypass the “Comments to the Author” section, enter your conflict of interest statement in the “Confidential to Editor” section, and submit your "Accept" recommendation.

Reviewer #1: All comments have been addressed

Reviewer #2: All comments have been addressed

2. Is the manuscript technically sound, and do the data support the conclusions?

Reviewer #1: Yes

Reviewer #2: (No Response)

3. Has the statistical analysis been performed appropriately and rigorously? 

Reviewer #1: Yes

Reviewer #2: (No Response)

4. Have the authors made all data underlying the findings in their manuscript fully available?

Reviewer #1: Yes

Reviewer #2: (No Response)

5. Is the manuscript presented in an intelligible fashion and written in standard English?

Reviewer #1: Yes

Reviewer #2: (No Response)

6. Review Comments to the Author

Reviewer #1: The authors have responded to all of this reviewer's comments. This reviewer is satisfied with the responses and the manuscript may be accepted.

Reviewer #2: (No Response)

7. PLOS authors have the option to publish the peer review history of their article (what does this mean?). If published, this will include your full peer review and any attached files.

Reviewer #1: No

Reviewer #2: No

---

## [Editor Report · Acceptance letter]

20 Aug 2020

PONE-D-20-11320R1 

Expression of EMP1, EMP2, and EMP3 in breast phyllodes tumors 

Dear Dr. Koo:

I'm pleased to inform you that your manuscript has been deemed suitable for publication in PLOS ONE. Congratulations! Your manuscript is now with our production department. 

Kind regards, 

on behalf of

Dr. Jung Weon Lee 

Academic Editor

PLOS ONE